# Looks Too Good To Be True:
# An Information-Theoretic Analysis of Hallucinations in Generative Restoration Models

**Regev Cohen**        **Idan Kligvasser**        **Ehud Rivlin**        **Daniel Freedman**

**Verily AI (Google Life Sciences), Israel**
regevcohen@google.com

## Abstract

The pursuit of high perceptual quality in image restoration has driven the development of revolutionary generative models, capable of producing results often visually indistinguishable from real data. However, as their perceptual quality continues to improve, these models also exhibit a growing tendency to generate hallucinations – realistic-looking details that do not exist in the ground truth images. Hallucinations in these models create uncertainty about their reliability, raising major concerns about their practical application. This paper investigates this phenomenon through the lens of information theory, revealing a fundamental tradeoff between uncertainty and perception. We rigorously analyze the relationship between these two factors, proving that the global minimal uncertainty in generative models grows in tandem with perception. In particular, we define the inherent uncertainty of the restoration problem and show that attaining perfect perceptual quality entails at least twice this uncertainty. Additionally, we establish a relation between distortion, uncertainty and perception, through which we prove the aforementioned uncertainly-perception tradeoff induces the well-known perception-distortion tradeoff. We demonstrate our theoretical findings through experiments with super-resolution and inpainting algorithms. This work uncovers fundamental limitations of generative models in achieving both high perceptual quality and reliable predictions for image restoration. Thus, we aim to raise awareness among practitioners about this inherent tradeoff, empowering them to make informed decisions and potentially prioritize safety over perceptual performance.

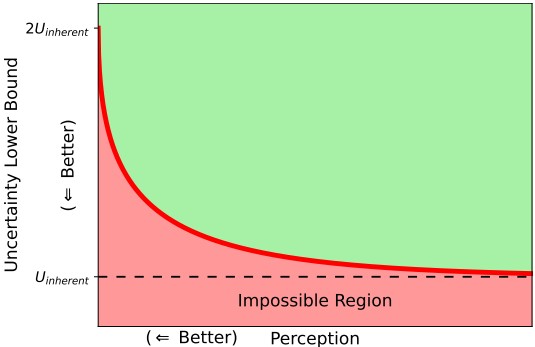

Figure 1: Illustration of Theorem 3. In restoration tasks, the minimal attainable uncertainty is lower bounded by a function that begins at the inherent uncertainty $U_{\text{Inherent}}$ of the problem (Definition 2) and graudally increases up to twice this value as the recovery approaches perfect perceptual quality.

38th Conference on Neural Information Processing Systems (NeurIPS 2024).

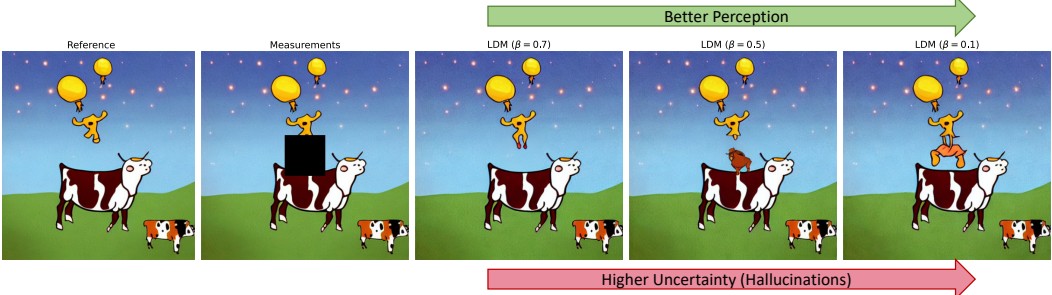

Figure 2: Image inpainting results. Algorithms are ordered from low to high perception (left to right). Note the corresponding increased hallucinations and distortion. See Section 5 for details.

# 1 Introduction

Restoration tasks and inverse problems impact many scientific and engineering disciplines, as well as healthcare, education, communication and art. Generative artificial intelligence [80, 38, 10] has transformed the field of inverse problems due to its unprecedented ability to infer missing information and restore corrupted data. In the realm of image restoration, the quest for high perceptual quality has led to a new generation of generative models, capable of producing outputs of remarkable realism, virtually indistinguishable from true images.

While powerful, growing empirical evidence indicates that generative models are susceptible to hallucinations [30], characterized by the generation of seemingly authentic content that deviates from the original input data, hindering applications where faithfulness is crucial. The root cause of hallucination lies in the ill-posed nature of restoration problems, where multiple possible solutions can explain the observed measurements, leading to uncertainty in the estimation process.

Concerns surrounding hallucinations have prompted the development of uncertainty quantification methods, designed to evaluate the reliability of generated outputs. These approaches offer crucial insights into the model's confidence in its predictions, empowering users to assess potential deviations from the original data and make informed decisions. Despite this progress, the relationship between achieving high perceptual quality and the extent of uncertainty remains an understudied area.

This paper establishes the theoretical relationship between uncertainty and perception, demonstrating through rigorous analysis that the global minimal uncertainty in generative models increases with the level of desired perceptual quality (see illustration in Figure 1). Leveraging information theory, we quantify uncertainty using the entropy of the recovery error [19], while we measure perceptual quality via conditional divergence between the distributions of the true and recovered images [58]. Our main contribution are as follows:

1. We introduce a definition for the inherent uncertainty $U_{\text{Inherent}}$ of an inverse problem, and formulate the uncertainty-perception (UP) function, seeking the minimal attainable uncertainty for a given perceptual index. We prove the UP function is globally lower-bounded by $U_{\text{Inherent}}$ (Theorem 1).

2. We prove a fundamental trade-off between uncertainty and perception under any underlying data distribution, restoration problem or model (Theorem 1). Specifically, the entropy power of the recovery error exhibits a lower bound inversely related to the Rényi divergence between the true and recovered image distributions (Theorem 3). This shows that perfect perceptual quality requires at least twice the inherent uncertainty $U_{\text{Inherent}}$.

3. We establish a relationship between uncertainty and mean squared error (MSE) distortion, demonstrating that the uncertainty-perception trade-off induces the well-known distortion-perception trade-off [14] (Theorem 4).

4. We empirically validate all theoretical findings through experiments on image super-resolution and inpainting (Section 5), covering a broad spectrum of recovery algorithms, diverse metrics and data distributions. Our experimental results for image inpainting are illustrated in Figure 2.

We aim to provide practitioners with a deeper understanding of the tradeoff between uncertainty and perceptual quality, allowing them to strategically navigate this balance and prioritize safety when deploying generative models in real-world, sensitive applications.

## 2 Related Work

Recent work in image restoration has made significant strides in both perceptual quality assessment and uncertainty quantification, largely independently. Below, we outline the main trends in research on these topics, laying the foundation for our framework.

**Perception Quantification** Perceptual quality in restoration tasks encompasses how humans perceive the output, considering visual fidelity, similarity to the original, and absence of artifacts. While traditional metrics like PSNR and SSIM [82] capture basic similarity, they miss finer details and higher-level structures. Learned metrics like LPIPS [87], VGG-loss [72], and DISTS [22] offer improvements but still operate on pixel or patch level, potentially overlooking holistic aspects. Recently, researchers have leveraged image-level embeddings from large vision models like DINO [17] and CLIP [62] to capture high-level similarity. Further advancements include HyperIQA [74] that leverages self-adaptive hyper networks to blindly assess image quality in the wild, while LIQE [88] and QAlign [84] utilize large language models to capture high-level semantic similarity and alignment between the restored and original images. Here, we follow previous works [58, 14, 31] and adopt a mathematical notion of perceptual quality defined as the divergence between probability densities.

**Uncertainty Quantification** Uncertainty quantification techniques can be broadly categorized into two main paradigms: Bayesian estimation and frequentist approaches. The Bayesian paradigm defines uncertainty by assuming a distribution over the model parameters and/or activation functions [1]. The most prevalent approach is Bayesian neural networks [52, 78, 34], which are stochastic models trained using Bayesian inference. To improve efficiency, approximation methods have been developed, including Monte Carlo dropout [24, 25], stochastic gradient Markov chain Monte Carlo [67, 18], Laplacian approximations [63] and variational inference [16, 51, 60]. Alternative Bayesian techniques encompass deep Gaussian processes [20], deep ensembles [7, 33], and deep Bayesian active learning [26]. In contrast to Bayesian methods, frequentist approaches operate assume fixed model parameters with no underlying distribution. Examples of such distribution-free techniques are model ensembles [44, 59], bootstrap [36, 2], interval regression [59, 37, 83] and quantile regression [27, 64].

An emerging approach in recent years is conformal prediction [3, 70], which leverages a labeled calibration dataset to convert point estimates into prediction regions. Conformal methods require no retraining, computationally efficient, and provide coverage guarantees in finite samples [49]. These works include conformalized quantile regression [64, 69, 6], conformal risk control [5, 8, 4], and semantic uncertainty intervals for generative adversarial networks [68]. The authors of [42] introduce the notion of conformal prediction masks, interpretable image masks with rigorous statistical guarantees for image restoration, highlighting regions of high uncertainty in the recovered images. Please see [75] for an extensive survey of distribution-free conformal prediction methods. A recent approach [11] introduces a principal uncertainty quantification method for image restoration that considers spatial relationships within the image to derive uncertainty intervals that are guaranteed to include the true unseen image with a user-defined confidence probabilities. While the above studies offer a variety of approaches for quantifying uncertainty, a rigours analysis of the relationship between uncertainty and perception remains underexplored in the context of image restoration.

**The Distortion-Perception Tradeoff** The most relevant studies to our research are the work on the distortion-uncertainty tradeoff [14] and its follow-ups [23, 15, 13]. A key finding in [14] establishes a convex tradeoff between perceptual quality and distortion in image restoration, applicable to any distortion measure and distribution. Moreover, perfect perceptual quality comes at the expense of no more than 3dB in PSNR. The work in [23] extends this, providing closed-form expressions for the tradeoff when MSE distortion and Wasserstein-2 distance are considered as distortion and perception measures respectively. In [58], it is shown that the Lipschitz constant of any deterministic estimator grows to infinity as it approaches perfect perception.

This work uniquely emphasizes *uncertainty* in image restoration, distinguishing it from distortion. While distortion measures how close a restored image is to the original, uncertainty quantifies the confidence in the restoration itself. This distinction is crucial for decision-making, as high uncertainty can hinder informed choices, complementing existing research on perceptual quality and robustness.

# 3  Problem Formulation

We adopt a Bayesian perspective to address inverse problems, wherein we seek to recover a random vector $X \in \mathbb{R}^d$ from its observations, represented by another random vector $Y = \mathcal{M}(X) \in \mathbb{R}^{d'}$. Here $\mathcal{M} : \mathbb{R}^d \to \mathbb{R}^{d'}$ is a non-invertible degradation function, implying $X$ cannot be perfectly recovered from $Y$. Formally:

**Definition 1.** *A degradation function $\mathcal{M}$ said to be invariable if, the conditional probability $p_{X|Y}(\cdot|y)$ is a Dirac delta function for almost every $y$ in the support of the distribution $p_Y$ of $Y$.*

The restoration process involves constructing a estimator $\hat{X} \in \mathbb{R}^d$ to estimate $X$ from $Y$, inducing conditional probability $p_{\hat{X}|Y}$. The estimation process forms a Markov chain $X \to Y \to \hat{X}$, implying that $X$ and $\hat{X}$ are statistically independent given $Y$.

In this paper, we analyze estimators $\hat{X}$ with respect to two performance criteria: perception and uncertainty. To assess perceptual quality, we follow a theoretical approach, similar to previous works [85, 14], and measure perception using conditional divergence[1] between $X$ and $\hat{X}$ defined as

$$D_v(X, \hat{X}|Y) \triangleq \mathbb{E}_{y \sim p_Y} \left[ D_v\big(p_{X|Y=y}, p_{\hat{X}|Y=y}\big) \right], \tag{1}$$

where $D_v$ stands for general divergence function. When an estimator attains a low value of the metric above, we say it exhibits high perceptual quality. When it comes to uncertainty, there are diverse practical methods to quantify it [28, 1]. However, for our analysis, we aim to identify a fundamental understanding of uncertainty. Therefore, we adopt the concept of entropy power from information theory, which assesses the statistical spread of a random variable. For the definition of entropy power and other relevant background, we refer the reader to Appendix B. Utilizing entropy power, we formally define the inherent uncertainty intrinsic to the restoration problem as follows

**Definition 2.** *The inherent uncertainty in estimating $X$ from $Y$ is defined as:*

$$U_{Inherent} \triangleq N(X|Y) = \frac{1}{2\pi e} e^{\frac{2}{d} h(X|Y)},$$

*where $h(X|Y)$ denotes the entropy of $X$ given $Y$.*

The inherent uncertainty quantifies the information irrevocably lost during observation, acting as a fundamental limit on the recovery of $X$ from $Y$, regardless of the estimation method. Notably, when the degradation process is invertible, this inherent uncertainty becomes zero $U_{Inherent} = 0$, reflecting the possibility of perfect recovery of $X$ with complete confidence.

We now turn our attention to the main focus of this paper, *the uncertainty-perception* (UP) function:

$$U(P) \triangleq \min_{p_{\hat{X}|Y}} \left\{ N(\hat{X} - X|Y) \ : \ D_v(X, \hat{X}|Y) \le P \right\}. \tag{2}$$

In essence, $U(P)$ represents the minimum uncertainty achievable by an estimator with perception quality of at least $P$, given the side information within the observation $Y$. In contrast to the perception-distortion function [14], the above objective prioritizes the information content of error signals over their mere energy, and its minimization promotes concentrated errors for robust and reliable predictions. The following example offers intuition into the typical behavior of this function.

**Example 1.** *Consider $Y = X + W$ where $X \sim \mathcal{N}(0, 1)$ and $W \sim \mathcal{N}(0, \sigma^2)$ are independent. Let the perception measure be the symmetric Kullback–Leibler (KL) divergence $D_{SKL}$ and assume stochastic estimators of the form $\hat{X} = \mathbb{E}[X|Y] + Z$ where $Z \sim \mathcal{N}(0, \sigma_z^2)$ is independent of $Y$. As derived in Appendix C, the UP function admits a closed form expression in this case, given by*

$$U(P) = N(X|Y)\left[1 + \left(P + 1 - \sqrt{(P+1)^2 - 1}\right)^2\right], \text{ where } N(X|Y) = \sigma^2/(1 + \sigma^2).$$

The above result, illustrated in Appendix C, demonstrates the minimal attainable uncertainty increases as the perception quality improves. Moreover, The above example suggests a structure for uncertainty-perception function $U(P)$, which fundamentally relies on the inherent uncertainty

---

[1]See Appendix A for a brief explanation of how conditional divergence relates to human perception.

$N(X|Y)$. Remarkably, the following section shows that this dependency generalizes beyond the specific example presented here, where its particular form is determined by the underlying distributions, along with the specific perception measure employed.

**Remark** One may consider the following alternative formulation

$$\tilde{U}(P) \triangleq \min_{p_{\hat{X}|Y}} \left\{ N(\hat{X} - X) \; : \; D_v(X, \hat{X}|Y) \leq P \right\}. \tag{3}$$

The alternative objective quantifies uncertainty as the entropy power of the error, independent of the side information $Y$. While potentially insightful, this approach may overestimate uncertainty since $N(\hat{X} - X|Y) \leq N(\hat{X} - X)$ where equality holds if and only if the error $\mathcal{E} = \hat{X} - X$ is independent of $Y$. Although further investigation is warranted, we hypothesize that the behavior of function (3) mirrors that of the UP function (2), which we examine in detail in the following section.

## 4 The Uncertainty-Perception Tradeoff

Thus far, we have formulated the uncertainty-perception function and elucidated its underlying rationale. We now proceed to derive its key properties, including a detailed analysis for the case where Rényi divergence serves as the measure of perceptual quality. Subsequently, we establish a direct link between the UP function and the well-known distortion-perception tradeoff. Finally, we demonstrate our theoretical findings through experiments on image super-resolution.

### 4.1 The Uncertainty-Perception Plane

The following theorem establishes general properties of the uncertainty-perception function, $U(P)$, irrespective of the specific distributions and divergence measures chosen.

**Theorem 1.** *The uncertainty-perception function $U(P)$ displays the following properties*

*1. Quasi-linearity (monotonically non-increasing and continuous):*

$$\min\left(U(P_1), U(P_2)\right) \leq U\left(\lambda P_1 + (1 - \lambda)P_2\right) \leq \max\left(U(P_1), U(P_2)\right), \; \forall \lambda \in [0, 1]$$

*2. Boundlessness:*

$$N(X|Y) \leq U(P) \leq 2N(X_G|Y),$$

*where $X_G$ is a zero-mean Gaussian random variable with covariance identical to $X$. The inherent uncertainty is upper bounded by $N(X_G|Y)$, which depends on the deviation of $X$ from Gaussianity.*

The theorem establishes a fundamental tradeoff between perceptual quality and uncertainty in image restoration, regardless of the specific divergence measure, data distributions, or restoration model employed. This tradeoff is fundamentally linked to the inherent uncertainty $N(X|Y)$ arising from the information loss during the observation process. Notably, the upper bound can be expressed as

$$N(X_G|Y) = N(X|Y)e^{\frac{2}{d}D_{KL}(X, X_G|Y)}. \tag{4}$$

This shows that as $X$ approaches Gaussianity, $N(X|Y)$ approaches $N(X_G|Y)$. However, concurrently, it implies in general higher values of $N(X|Y)$ due to Lemma 1 of Appendix B. This finding yields a surprising insight: for multivariate Gaussian distributions, perfect perceptual quality comes at the expense of exactly twice the inherent uncertainty of the problem.

Next, we show that for a fixed perceptual index $P$, the optimal algorithms lie on the boundary of the constraint set. This facilitates the optimization, as it restricts the search space to the boundary points.

**Theorem 2.** *Assume $D_v(X, \hat{X}|Y)$ is convex in its second argument. Then, for any $P \geq 0$, the minimum is attained on the boundary where $D_v(X, \hat{X}|Y) = P$.*

Note that the assumption of the convexity of $\mathcal{D}_v$ in its second argument is not a restrictive condition. In fact, most widely-used divergence functions, notably all $f$-divergences (such as KL divergence, total variation distance, Hellinger distance, and Chi-square divergence), exhibit this property.

While the above theorems describe important characteristics of the uncertainty-perception function, additional assumptions are needed to gain deeper insights. Therefore, we now focus on Rényi divergence as our perception measure. Rényi divergence is a versatile family of divergence functions parameterized by an order $0 \leq r$, encompassing the well-known KL divergence as a special case when $r = 1$. This divergence plays a critical role in in analyzing Bayesian estimators and numerous information theory calculations [79]. Importantly, it is also closely related to other distance metrics used in probability and statistics, such as the Wasserstein and Hellinger distances. Focusing on the case where $r = 1/2$, we arrive at:

$$U(P) = \min_{p_{\hat{X}|Y}} \left\{ N(\hat{X} - X|Y) \; : \; D_{1/2}(X, \hat{X}|Y) \leq P \right\}. \tag{5}$$

While we set $r = 1/2$ to facilitate our derivations, it is important to note that all orders $r \in (0, 1)$ are equivalent (see Appendix B). Consequently, given this equivalence and the close relationship between Rényi divergence and other metrics, analyzing the specific formulation provided by (5) may yield valuable insights applicable to a wide range of divergence measures. The following theorem provides lower and upper bounds for the UP function.

**Theorem 3.** *The uncertainty-perception function is confined to the following region*

$$\eta(P) \cdot N(X|Y) \; \leq \; U(P) \; \leq \; \eta(P) \cdot N(X_G|Y)$$

*where $1 \leq \eta(P) \leq 2$ is a convex function w.r.t the perception index and is given by*

$$\eta(P) = \left( 2e^{\frac{2P}{d}} - \sqrt{(2e^{\frac{2P}{d}} - 1)^2 - 1} \right).$$

Noteworthy, Theorem 3 holds true regardless of the underlying distributions of $X$ and $Y$, thereby providing a universal characterization of the UP function in terms of perception. Furthermore, as depicted in Figure 3, Theorem 3 gives rise to the uncertainty-perception plane, which divides the space into three distinct regions:

1. Impossible region, where no estimator can reach.
2. Optimal region, encompassing all estimators that are optimal according to (5).
3. Suboptimal region of estimators which exhibit overly high uncertainty.

The existence of an impossible region highlights the uncertainty-perception tradeoff, proving no estimator can achieve both high perception and low uncertainty simultaneously. This finding underscores the importance of practitioners being aware of this tradeoff, enabling them to make informed decisions when prioritizing between perceptual quality and uncertainty in their applications. The uncertainty-perception plane could serve as a valuable framework for evaluating estimator performance in this context. While not a comprehensive metric, it may offer insights into areas where improvements can be made, guiding practitioners towards estimators that strike a more desirable balance between perception and uncertainty. For certain estimators residing in the suboptimal region, it may be possible to achieve lower uncertainty without sacrificing perceptual quality. Thus, we believe that our proposed uncertainty-perception plane can serve as a valuable starting point for further research and practical applications, ultimately leading to the development of safer and reliable image restoration algorithms.

Next, we analyze how the dimensionality of the underlying data affects the uncertainty-perception tradeoff. To achieve this, we extend the function $\eta(P)$ to include a dimension parameter $d$, denoted as $\eta(P; d)$. As shown in Fig. 4, $\eta(P; d)$ exhibits a rapid incline as perception improves and it attain higher values in higher dimensions. This observation suggests that in high-dimensional settings, the uncertainty-perception tradeoff becomes more severe, implying that any marginal improvement in perception for an algorithm is accompanied by a dramatic increase in uncertainty.

Finally, we conjecture that the general form of the tradeoff, given by the inequality in Theorem 3, holds for different divergence measures, with the specific form of $\eta(P)$ capturing the nuances of each chosen measure. For instance, considering the Hellinger distance as our perception measure, we obtain the same inequality as in Theorem 3 but with $\eta(P)$ defined for $0 \leq P \leq 1$ as[2]

$$\eta_{\text{Hellinger}}(P) = \frac{2}{(1-P)^{4/d}} - \sqrt{\left( \frac{2}{(1-P)^{4/d}} - 1 \right)^2 - 1}. \tag{6}$$

---

[2]The case of $P = 1$ is obtained by taking the limit $\lim_{P \to 1} \eta(P) = 1$.

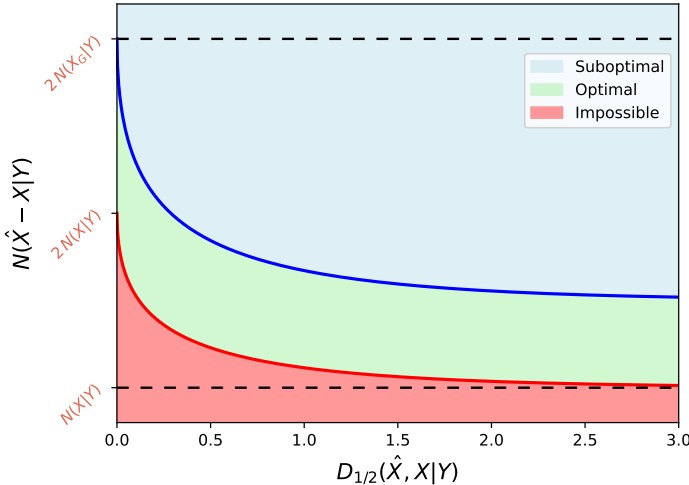

Figure 3: The uncertainty-perception plane (Theorem 3). The impossible region demonstrates the inherent tradeoff between perception and uncertainty, while other regions may guide practitioners toward estimators that better balance the two factors, highlighting potential areas for improvement.

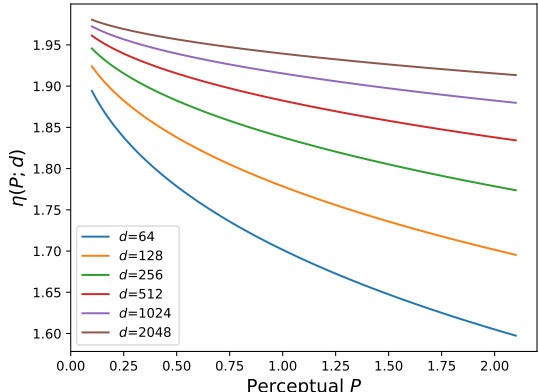

Figure 4: Impact of dimensionality, as revealed in Theorem 3, demonstrates that the uncertainty-perception tradeoff intensifies in higher dimensions. This implies that even minor improvements in perceptual quality for an algorithm may come at the cost of a significant increase in uncertainty.

## 4.2 Revisiting the Distortion-Perception Tradeoff

Having established the uncertainty-perception tradeoff and its characteristics, we now broaden our analysis to estimation distortion, particularly the mean squared-error. A well-known result in estimation theory states that for any random variable $X$ and for any estimator $\hat{X}$ based upon side information $Y$, the following holds true [19]:

$$\mathbb{E}\left[||\hat{X} - X||^2\right] \geq \frac{1}{2\pi e} e^{2h(X|Y)}. \tag{7}$$

This inequality, related to the uncertainty principle, serves as a fundamental limit to the minimal MSE achieved by any estimator. However, it does not consider the estimation uncertainty of $\hat{X}$ as the right hand side is independent of $\hat{X}$. Thus, we extend the above in the following theorem.

**Theorem 4.** *For any random variable $X$, observation $Y$ and unbiased estimator $\hat{X}$, it holds that*

$$\frac{1}{d}\mathbb{E}\left[||\hat{X} - X||^2\right] \geq N\left(\hat{X} - X|Y\right).$$

Notice that for any estimator $\hat{X}$ we have $N(\hat{X} - X|Y) \geq N(X|Y)$, implying

$$\frac{1}{d}\mathbb{E}[||\hat{X} - X||^2] \geq N(X|Y) = \frac{1}{2\pi e} e^{\frac{2}{d} h(X|Y)}. \tag{8}$$

The above result aligns with equation (7), demonstrating that Theorem 4 serves as a generalization of inequality (7), incorporating the uncertainty associated with the estimation. Furthermore, by viewing the estimator $\hat{X}$ as a function of perception index $P$, we arrive at the next corollary.

**Corollary 1.** *Define the following distortion-perception function*

$$D(P) \triangleq \min_{p_{\hat{X}|Y}} \left\{ \frac{1}{d} \mathbb{E}\left[||\hat{X} - X||^2\right] : D_v(X, \hat{X}|Y) \leq P \right\}.$$

*Then, for any perceptual index $P$, we have $D(P) \geq U(P)$.*

As uncertainty increases with improving perception, the corollary implies that distortion also increases. Thus, when utilizing MSE as a measure of distortion, the uncertainty-perception tradeoff induces a distortion-perception tradeoff [14], offering a novel interpretation of this well-known phenomenon.

## 5    Experiments

**Setup.** Our theoretical framework is grounded in empirical observations, leading us to validate our findings through experiments on common benchmark tasks: image super-resolution and inpainting. We analyze performance through the lens of uncertainty, alongside established measures of perceptual quality and distortion. To assess perceptual quality, we employ state-of-the-art metrics including HyperIQA [74], LIQE [88] and Q-ALIGN [84]. Distortion is evaluated using traditional measures: MSE, peak signal-to-noise ratio (PSNR), and structural similarity index (SSIM) [82]. Accurately estimating entropy in high-dimensional spaces presents significant challenges [46]; hence, we utilize an upper bound for uncertainty, $N(\hat{X}_G - X_G|Y)$, as detailed in Appendix F. This practical alternative simplifies computation to calculating the geometric mean of the singular values of the error covariance.

For super-resolution, we utilize the BSD100 benchmark dataset [55], aiming to predict a high-resolution image from its low-resolution counterpart obtained via $4\times$ bicubic downsampling. Our evaluation spans a diverse range of recovery algorithms, including EDSR [50], ESRGAN [81], SinGAN [71], SANGAN [39], DIP [77], SRResNet/SRGAN variants [47], EnhanceNet [66], and Latent Diffusion Models (LDMs) with parameter $\beta \in [0, 1]$ [65], where $\beta = 0$ recovers DDIM [32] and $\beta = 1$ recovers DDPM [73]. In the context of image inpainting, we leverage the SeeTrue dataset [86], an image-text alignment benchmark known for its diverse collection of real and synthetic text-image pairs. Here, we focus our analysis on diffusion models due to their state-of-the-art performance and growing popularity in the field.

**Results.** Figure 5 presents our super-resolution analysis. As observed in the top row, across various perceptual measures, an unattainable blank region exists in the lower right corner, indicating that no model simultaneously achieves both low uncertainty and high perceptual quality. Furthermore, an anti-correlation emerges near this region, where modest improvements in perceptual quality translate to dramatic increases in uncertainty. This observation suggests the existence of a tradeoff between uncertainty and perception. Additionally, the bottom row showcases a strong relationship between uncertainty and distortion across diverse measures, demonstrating that any increase in uncertainty leads to a significant rise in distortion.[3] Figure 6 displays similar trends for image inpainting, consistent with our super-resolution analysis and reinforcing the validity of our findings across diverse restoration tasks and data distributions. This is further visualized in Figure 2, which presents outputs from selected algorithms ordered by perceptual quality. The results clearly demonstrate an increase in hallucination (uncertainty) and distortion with increasing perceptual quality. Finally, Appendix H presents additional results obtained via direct estimation of statistics in high dimensions, further supporting our theoretical analysis.

## 6    Conclusion

This study established the uncertainty-perception tradeoff in generative restoration, demonstrating that high perceptual quality leads to increased hallucination (uncertainty), particularly in high dimensions. We characterized this tradeoff and its fundamental relation to the inherent uncertainty of the problem,

---

[3]Note that MSE is a measure of distortion, whereas PSNR and SSIM are measures of inverse distortion; this accounts for the negative slope in the first two figures, and the positive slope in the third.

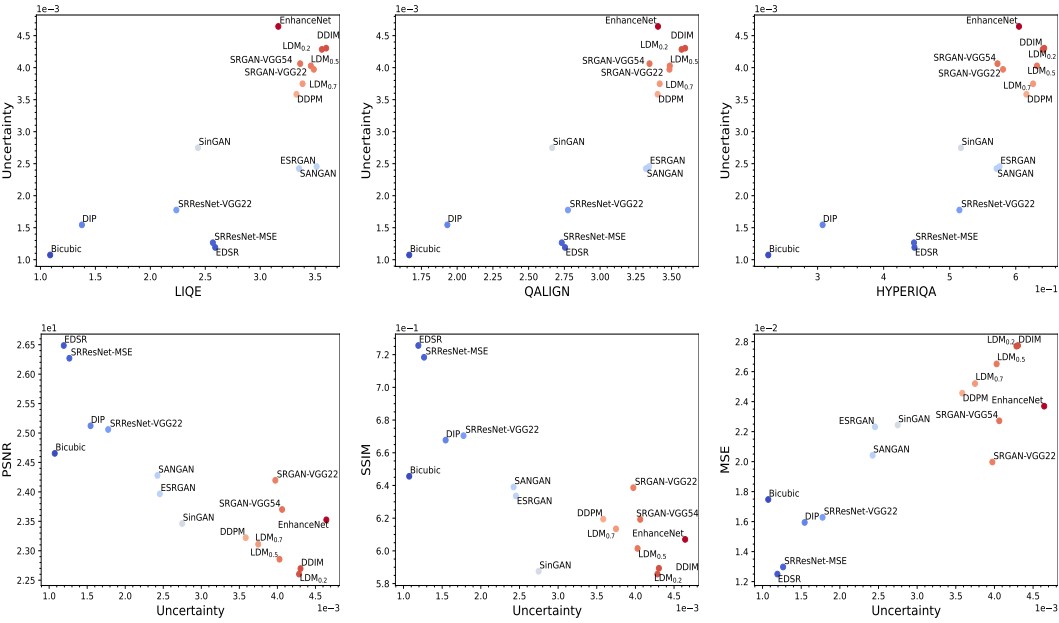

Figure 5: Evaluation of SR algorithms. Top: Uncertainty-perception plane showing the tradeoff between perceptual quality and uncertainty (y-axis) for various perceptual measures. Bottom: Uncertainty-distortion plane showing the relationship between uncertainty and various distortion measures. Axis placement differs in the two rows to highlight the distinct roles of uncertainty.

introducing the uncertainty-perception plane which may guide practitioners in understanding estimator performance. By extending our analysis to MSE distortion, we showed that the distortion-perception tradeoff emerges as a direct consequence of the uncertainty-perception tradeoff. Experimental results confirmed our theoretical findings, highlighting the importance of this tradeoff in image restoration.

## 7 Limitations

Our analysis is grounded in the theoretical framework of entropy as a measure of uncertainty. Information theory offers a powerful framework for quantifying uncertainty and dependencies in data, handling multivariate and heterogeneous data types, and capturing complex patterns. However, its wider adoption has been limited by the challenge of estimating information-theoretic measures in high dimensions. The curse of dimensionality makes accurate density estimation infeasible [12, 48], leading many to rely on simpler second-order statistics.

The development of practical tools for estimating statistics in high-dimensional data remains an active area of research [76]. While initial approaches assumed exponential family distributions (e.g., Gaussian) for tractable calculations [57], their performance degrades for long-tailed distributions. Non-parametric methods like binning strategies, including KDE and kNN estimators [61, 40, 29], offer more flexibility but are data-dependent and sensitive to parameter choices. Alternative approaches involve ensemble estimation [43] or von Mises Expansions [35], the distributional analog of the Taylor expansion. Rotation-Based Iterative Gaussianization [46] presents a promising direction by transforming data into a multivariate Gaussian domain, simplifying density estimation. However, its application to images has been limited to small patches due to the computational challenges of learning rotations based on principal or independent component analysis. A recent extension addresses this by utilizing convolutional rotations, enabling efficient processing of entire images [45].

While accurately estimating high-dimensional entropy remains an active research area, Section 5 utilizes a tractable upper bound. This alternative calls for further investigation into its potential for quantifying uncertainty and analyzing algorithm performance. Moreover, incorporating this bound into the design of new algorithms could enable explicit control over the uncertainty-perception trade-off, potentially leading to more reliable solutions.

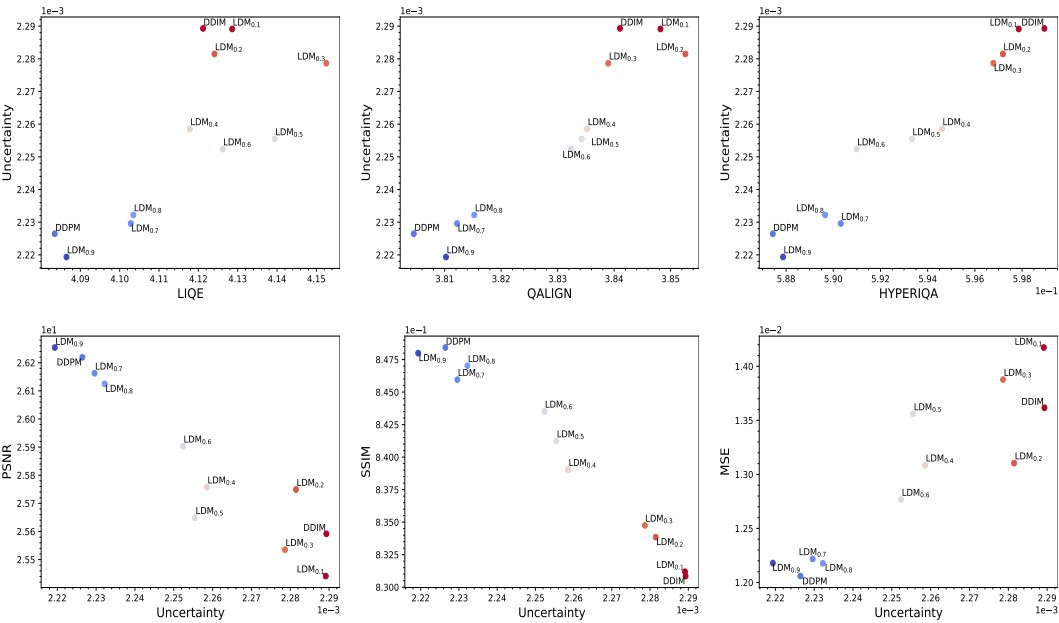

Figure 6: Evaluation of LDMs on image inpainting, highlighting the trade-off between uncertainty and perceptual quality (top) and the uncertainty-distortion relationship (bottom). No model achieves both low uncertainty and high perceptual quality, with higher uncertainty generally leading to increased distortion. Differing axis placements emphasize the distinct roles of uncertainty.

Lastly, we focused our empirical validation on image super-resolution and inpainting, two benchmark problems in image restoration. Our analysis, however, applies to any restoration task with non-invertible degradation. Hence, expanding the experiments to additional image-to-image tasks and domains such as audio, video, and text may reveal broader implications and applications of our work.

# 8 Broader Impact

Our work revealing a fundamental tradeoff between uncertainty and perception in image restoration carries significant societal impact. Developers across various fields, including healthcare and autonomous systems, often integrate cutting-edge models into their applications, prioritizing state-of-the-art performance and perceptual quality. However, our work aims to highlight a crucial factor often overlooked: the inherent tradeoff between uncertainty and perception. By raising awareness of this tradeoff, we empower developers to make informed decisions that prioritize safety and reliability over purely perceptual enhancements. For instance, in healthcare, potential restoration algorithms can be evaluated by plotting them on the uncertainty-perception plane, facilitating the identification of methods that strike the optimal balance for specific clinical needs. Furthermore, by understanding this inherent trade-off, practitioners can consider trading performance for better safety and resilience against potential misuse and misinterpretations.

While primarily theoretical, our analysis yields a practical measure of uncertainty (or entropy), used in our experiments to visually and quantitatively illustrate our findings. This tractable uncertainty measure, or any differentiable alternative, can be incorporated into a loss function during the training of generative models like GANs or as an optimization objective to guide the reverse process in diffusion models. This approach enables the development of algorithms that explicitly optimize for the tradeoff between uncertainty and perception.

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

## A  Conditional Divergence and Human Perception

In our context, perception is defined as the probability $p_{\text{success}}$ of a human observer successfully distinguishing between a pair of natural and degraded images, drawn from $p_{X,Y}$), and a pair of restored and degraded images drawn from $p_{\hat{X},Y}$). From a Bayesian perspective, the optimal decision rule maximizing $p_{\text{success}}$ yields ([56] Section 2):

$$p_{\text{success}} = \frac{1}{2} + \frac{1}{2} D_{\text{TV}}(p_{X,Y}, p_{\hat{X},Y})$$

where $D_{\text{TV}}(p_{X,Y}, p_{\hat{X},Y})$ is the total-variation (TV) distance. When $D(p_{X,Y}, p_{\hat{X},Y}) = 0$, the two pairs are indistinguishable ($p_{\text{success}} = 0.5$), implying perfect perception quality. We generalize this beyond the total-variation (TV) distance to any conditional divergence, recognizing that the divergence that best relates to human perception remains an open question.

## B  Information-Theory Preliminaries

To make the paper self-contained, we briefly overview the essential definitions and results in information-theory. Let $X$, $Y$ and $Z$ be continuous random variables with probability density functions $p_X(x)$, $p_Y(y)$ and $p_Z(z)$ respectively. The space of probability density functions is denoted by $\Omega$. We assume the quantities described below, which involve integrals, are well-defined and finite.

**Definition 3** (**Entropy**). *The differential entropy of $X$, whose support is a set $S_x$, is defined by*

$$h(X) \triangleq - \int_{S_X} p_X(x) \log p_X(x) dx.$$

**Definition 4** (**Rényi Entropy**). *The Rényi entropy of order $r \geq 0$ of $X$ is defined by*

$$h_r(X) \triangleq \frac{1}{1-r} \log \int p_X^r(x) dx.$$

*The above quantity generalizes various notions of entropy, including Hartley entropy, collision entropy, and min-entropy. In particular, for $r = 1$ we have*

$$h_1(X) \triangleq \lim_{r \to 1} h_r(X) = h(X).$$

**Definition 5** (**Entropy Power**). *Let be $h(X)$ be the differential entropy of $X \in \mathbb{R}^d$. Then, the entropy Power of $X$ is given by*

$$N(X) \triangleq \frac{1}{2\pi e} e^{\frac{2}{d} h(X)}.$$

**Definition 6** (**Divergence**). *A statistical divergence is any function $D_v : \Omega \times \Omega \to \mathbb{R}^+$ which satisfies the following conditions for all $p, q \in \Omega$:*

1.  $D_v(p, q) \geq 0$.

2.  $D_v(p, q) = 0$ iff $p = q$ almost everywhere.

Table 1: Formulas for Multivariate Gaussian Distribution

| Distribution | Quantity | Closed-Form Expression |
|---|---|---|
| $X \sim \mathcal{N}(\mu_x, \Sigma_x)$ | $h(X)$ | $\frac{1}{2} \ln\{(2\pi e)^d \lvert \Sigma_x \rvert\}$. |
| $X \sim \mathcal{N}(\mu_x, \Sigma_x)$ | $N(X)$ | $\lvert \Sigma_x \rvert^{1/n}$. |
| $X \sim \mathcal{N}(\mu_x, \Sigma_x)$ | $h_{\frac{1}{2}}(X)$ | $\frac{1}{2} \ln\{(8\pi)^d \lvert \Sigma_x \rvert\}$. |
| $X \sim \mathcal{N}(\mu_x, \Sigma_x),$ $Y \sim \mathcal{N}(\mu_y, \Sigma_y)$ | $D_{1/2}(X, Y)$ | $\frac{1}{4}(\mu_x - \mu_y)^T \left( \frac{\Sigma_x + \Sigma_y}{2} \right)^{-1} (\mu_x - \mu_y) + \ln\left( \frac{\left\lvert \frac{\Sigma_x + \Sigma_y}{2} \right\rvert}{\sqrt{\lvert \Sigma_x \rvert \lvert \Sigma_y \rvert}} \right)$. |

**Definition 7 (Rényi Divergence).** *The Rényi divergence of order $r \geq 0$ between $p_X$ and $p_Y$ is*

$$D_r(X, Y) \triangleq \frac{1}{r-1} \log \int p_X^r(x) p_Y^{1-r}(x) dx.$$

*The above establishes a spectrum of divergence measures, generalising the Kullback–Leibler divergence as $D_1(X, Y) = D_{KL}(X, Y)$. Furthermore, it is important to note that all orders $r \in (0, 1)$ are equivalent [79], since*

$$\frac{r}{t} \frac{1-t}{1-r} D_t(\cdot, \cdot) \leq D_r(\cdot, \cdot) \leq D_t(\cdot, \cdot), \ \forall 0 < r \leq t < 1. \tag{9}$$

**Definition 8 (Conditioning).** *Consider the joint probability $p_{XY}$ and the conditional probabilities $p_{X|Y}(x|y)$ and $p_{Z|Y}(z|y)$. The conditional differential entropy of $X \in \mathbb{R}^d$ given $Y$ is defined as*

$$h(X|Y) \triangleq - \int_{S_{XY}} p_{XY}(x, y) \log p_{X|Y}(x|y) dx dy$$
$$= \mathbb{E}_{y \sim p_Y} [h(X|Y = y)]$$

*where $S_{XY}$ is the support set of $p_{XY}$. Then, the conditional entropy power of $X$ given $Y$ is*

$$N(X|Y) = \frac{1}{2\pi e} e^{\frac{2}{d} h(X|Y)}.$$

*Similarly, the conditional divergence between $X$ and $Z$ given $Y$ is defined as*

$$D_v(X, Z|Y) \triangleq \mathbb{E}_{y \sim p_Y} [D_v(X|Y = y, Z|Y = y)].$$

*For example, the conditional Rényi divergence is given by*

$$D_r(X, Z|Y) \triangleq$$
$$\int \left( \frac{1}{r-1} \log \int p_{X|Y}^r(x|y) p_{Z|Y}^{1-r}(x|y) dx \right) p_Y \, dy.$$

Table 1 summarizes closed-form expressions for several quantities relevant to the multivariate Gaussian distribution. Below we present two fundamental results that form the basis of our analysis.

**Lemma 1 (Maximum Entropy Principle [19]).** *Let $X \in \mathbb{R}^d$ be a continuous random variable with zero mean and covariance $\Sigma_x$. Define $X_G \sim \mathcal{N}(0, \Sigma_x)$ to be a Gaussian random variable, independent of $X$, with the identical covariance matrix $\Sigma_{x_G} = \Sigma_x$. Then,*

$$h(X) \leq h(X_G),$$
$$N(X) \leq N(X_G) = |\Sigma_x|^{1/d}.$$

**Lemma 2 (Entropy Power Inequality [53]).** *Let $X$ and $Y$ be independent continuous random variables. Then, the following inequality holds*

$$N(X) + N(Y) \leq N(X + Y),$$

*where equality holds iff $X$ and $Y$ are multivariate Gaussian random variables with proportional covariance matrices. Equivalently, let $X_g$ and $Y_g$ be defined as independent, isotropic multivariate Gaussian random variables satisfying $h(X_g) = h(X)$ and $h(Y_g) = h(Y)$. Then,*

$$h(X) + h(Y) = h(X_g) + h(Y_g) = h(X_g + Y_g) \leq h(X + Y).$$

## C   Derivation of Example 1

Since $\hat{X} = \mathbb{E}[X|Y] + Z$, then $\hat{X}|Y \sim \mathcal{N}(\mathbb{E}[X|Y], \sigma_z^2)$. Moreover, $X|Y \sim \mathcal{N}(\mathbb{E}[X|Y], \sigma_q^2)$ where $\sigma_q^2 = \frac{\sigma^2}{1+\sigma^2}$. Thus, the conditional error entropy is given by $N(\hat{X} - X|Y) = \sigma_q^2 + \sigma_z^2$ and the symmetric KL divergence is $D_{SKL}(X, \hat{X}|Y) = \frac{\sigma_q^2 + \sigma_z^2}{2\sigma_z \sigma_q} - 1$, leading the following problem

$$U(P) = \min_{\sigma_z} \left\{ \sigma_q^2 + \sigma_z^2 \ : \ \frac{\sigma_q^2 + \sigma_z^2}{2\sigma_z \sigma_q} - 1 \leq P \right\}. \tag{10}$$

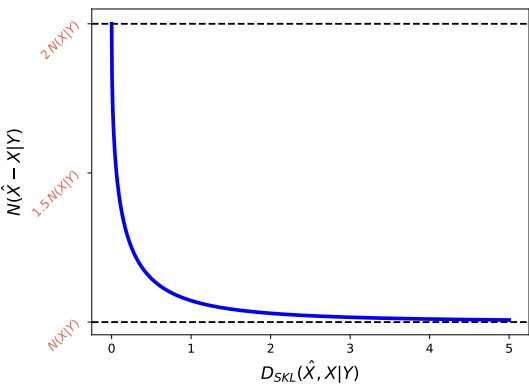

Figure 7: The Uncertainty-Perception function for Example 1. As perception quality improves, the minimal achievable uncertainty increases, suggesting a tradeoff governed by the inherent uncertainty.

Therefore, we seek the minimal value of $\sigma_z$ that satisfies the constraint. Note that the minimal value is attained at the boundary of the constraint set, where the inequality becomes an equality

$$\frac{\sigma_q^2 + \sigma_z^2}{2\sigma_z\sigma_q} - 1 = P \;\Rightarrow\; \sigma_z^2 - 2\sigma_q(P+1)\sigma_z + \sigma_q^2 = 0. \tag{11}$$

The solution to the aforementioned quadratic problem is $\sigma_z^* = \sigma_q\left(P+1-\sqrt{(P+1)^2-1}\right)$. Substituting the later into the objective function, we obtain

$$U(P) = \sigma_q^2\left[1+\left(P+1-\sqrt{(P+1)^2-1}\right)^2\right]. \tag{12}$$

Finally, the entropy power of an univariate Gaussian distribution equals its variance $\sigma_q^2 = N(X|Y)$. Figure 7 visualizes the resulting uncertainty-perception tradeoff.

## D  Proof of Theorem 1

First, the constraint $\mathcal{C}(P) \triangleq \{\hat{X} \,:\, D_v(X, \hat{X}|Y) \leq P\}$ defines a compact set which is continuous in $P$. Hence, by the Maximum Theorem [19], $U(P)$ is continuous. In addition, $U(P)$ is the minimal error entropy power obtained over a constraint set whose size does not decrease with $P$, thus, $U(P)$ is non-increasing in $P$. Any continuous non-increasing function is quasi-linear. For the lower bound consider the case where $P = \infty$, leading to the following unconstrained problem

$$U(\infty) \triangleq \min_{p_{\hat{X}|Y}} N(\hat{X}-X|Y). \tag{13}$$

For any $P \geq 0$ it holds that $U(\infty) \leq U(P)$, and by Lemma 2 we have

$$N(X|Y) + \min_{p_{\hat{X}|Y}} N(\hat{X}|Y) \leq U(\infty). \tag{14}$$

Since $\min_{p_{\hat{X}|Y}} N(\hat{X}|Y) \geq 0$ we obtain

$$\forall P \geq 0: \quad N(X|Y) \leq U(P). \tag{15}$$

Next, we have $U(P) \leq U(0) = N(\hat{X}_0 - X|Y)$ where $p_{\hat{X}_0|Y} = p_{X|Y}$. Define $V \triangleq \hat{X}_0 - X$, then $\Sigma_{v|y} = \Sigma_{\hat{x}|y} + \Sigma_{x|y} = 2\Sigma_{x|y}$ where we use that $X$ and $\hat{X}$ are independent given $Y$. Thus,

$$U(0) = N(V|Y) \leq N(V_G|Y) = \left|\Sigma_{v|y}\right|^{1/d} = \left|2\Sigma_{x|y}\right|^{1/d} = 2\left|\Sigma_{x|y}\right|^{1/d} = 2N(X_G|Y), \tag{16}$$

where the first inequality is due to Lemma 1. Finally, for any $P \geq 0$ it holds that $U(P) \leq U(0)$ which implies $U(0) \leq 2N(X_G|Y)$, completing the proof.

# E    Proof of Theorem 2

Assuming $D_v(X, \hat{X}|Y)$ is convex in its second argument, the constraint represent a compact, convex set. Moreover, $h(\hat{X} - X|Y)$ is strictly-concave w.r.t $p_{\hat{X}|Y}$ as a composition of a linear function (convolution) with a strictly-concave function (entropy). Therefore, we minimize a log-concave function over a convex domain and thus the global minimum is attained on the set boundary where $D_v(X, \hat{X}|Y) = P$.

# F    Proof of Theorem 3

We begin with applying Lemma 1 and Lemma 2 to bound the objective function as follows

$$N(\hat{X}_g|Y) + N(X_g|Y) = N(\hat{X}_g - X_g|Y) \le N(\hat{X} - X|Y) \le N(\hat{X}_G - X_G|Y). \quad (17)$$

Note that the bounds are tight as the upper bound is attained when $\hat{X}|Y$ and $X|Y$ are multivariate Gaussian random variables, while the lower bound is attained if we further assume they are isotropic. Thus, we can bound the uncertainty-perception function as follows

$$U_g(P) \le U(P) \le U_G(P) \quad (18)$$

where we define

$$
\begin{aligned}
U_g(P) &\triangleq \min_{p_{\hat{X}_g|Y}} \left\{ N(\hat{X}_g|Y) + N(X_g|Y) \; : \; D_{1/2}(X_g, \hat{X}_g|Y) \le P \right\}, \\
U_G(P) &\triangleq \min_{p_{\hat{X}_G|Y}} \left\{ N(\hat{X}_G - X_G|Y) \; : \; D_{1/2}(X_G, \hat{X}_G|Y) \le P \right\}.
\end{aligned}
\quad (19)
$$

The above quantities can be expressed in closed form. We start with minimization problem of the upper bound which can be written as

$$U_G(P) = \min_{p_{\hat{X}_G|Y}} \left\{ \frac{1}{2\pi e} e^{\frac{2}{d} \mathbb{E}[h(\hat{X}_G - X_G|Y=y)]} \; : \; \mathbb{E}\left[ D_{1/2}(X_G, \hat{X}_G|Y = y) \right] \le P \right\}, \quad (20)$$

where the expectation is over $y \sim Y$. Substituting the expressions for $h(X_G - X_G|Y = y)$ and $D_{1/2}(X_G, \hat{X}_G|Y = y)$, we get

$$U_G(P) = \min_{\{\Sigma_{\hat{x}|y}\}} \left\{ \frac{1}{2\pi e} e^{\frac{2}{d} \mathbb{E}\left[ \frac{1}{2} \log \left\{ (2\pi e)^d |\Sigma_{\hat{x}|y} + \Sigma_{x|y}| \right\} \right]} \; : \; \mathbb{E}\left[ \log \frac{\left| \left( \Sigma_{\hat{x}|y} + \Sigma_{x|y} \right)/2 \right|}{\sqrt{|\Sigma_{\hat{x}|y}| \, |\Sigma_{x|y}|}} \right] \le P \right\}. \quad (21)$$

Notice the optimization is with respect to the covariance matrices $\{\Sigma_{\hat{x}|y}\}$. Simplifying the above, we can equivalently solve the following minimization

$$\min_{\{\Sigma_{\hat{x}|y}\}} \mathbb{E}\left[ \log \left| \Sigma_{\hat{x}|y} + \Sigma_{x|y} \right| \right] \quad \text{s.t.} \quad \mathbb{E}\left[ \log \frac{\left| \left( \Sigma_{\hat{x}|y} + \Sigma_{x|y} \right)/2 \right|}{\sqrt{|\Sigma_{\hat{x}|y}| \, |\Sigma_{x|y}|}} \right] \le P. \quad (22)$$

The solution of a constrained optimization problem can be found by minimization the Lagrangian

$$L\left( \{\Sigma_{\hat{x}|y}\}, \lambda \right) \triangleq \mathbb{E}\left[ \log \left| \Sigma_{\hat{x}|y} + \Sigma_{x|y} \right| \right] + \lambda \left( \mathbb{E}\left[ \log \frac{\left| \left( \Sigma_{\hat{x}|y} + \Sigma_{x|y} \right)/2 \right|}{\sqrt{|\Sigma_{\hat{x}|y}| \, |\Sigma_{x|y}|}} \right] - P \right). \quad (23)$$

Since expectation is a linear operation and using that $P = \mathbb{E}[P]$, we rewrite the above as

$$L\left( \{\Sigma_{\hat{x}|y}\}, \lambda \right) = \mathbb{E}\left[ \log \left| \Sigma_{\hat{x}|y} + \Sigma_{x|y} \right| + \lambda \left( \log \frac{\left| \left( \Sigma_{\hat{x}|y} + \Sigma_{x|y} \right)/2 \right|}{\sqrt{|\Sigma_{\hat{x}|y}| \, |\Sigma_{x|y}|}} - P \right) \right]. \quad (24)$$

The expression within the expectation can be written as

$$\log \left| \Sigma_{\hat{x}|y} + \Sigma_{x|y} \right| + \lambda \left( \log \left| \left( \Sigma_{\hat{x}|y} + \Sigma_{x|y} \right)/2 \right| - \frac{1}{2} \log \left| \Sigma_{\hat{x}|y} \right| - \frac{1}{2} \log \left| \Sigma_{x|y} \right| - P \right). \quad (25)$$

Next, according to KKT conditions the solutions should satisfy $\frac{\partial L}{\partial \Sigma_{\hat{x}|y}} = 0$. Using the linearity of the expectation and differentiating (25) w.r.t $\Sigma_{\hat{x}|y}$ we obtain

$$\left(\Sigma_{\hat{x}|y} + \Sigma_{x|y}\right)^{-1} + \lambda \left( \left(\Sigma_{\hat{x}|y} + \Sigma_{x|y}\right)^{-1} - \frac{1}{2}\Sigma_{\hat{x}|y}^{-1}\right) = 0 \tag{26}$$

Multiplying both sides by $\left(\Sigma_{\hat{x}|y} + \Sigma_{x|y}\right)$, we have

$$I + \lambda I - \frac{\lambda}{2}I - \frac{\lambda}{2}\Sigma_{x|y}\Sigma_{\hat{x}|y}^{-1} = 0$$
$$\Rightarrow (1 + \frac{\lambda}{2})I = \frac{\lambda}{2}\Sigma_{x|y}\Sigma_{\hat{x}|y}^{-1}$$
$$\Rightarrow (\lambda + 2)\Sigma_{\hat{x}|y} = \lambda\Sigma_{x|y} \tag{27}$$
$$\Rightarrow \Sigma_{\hat{x}|y} = \frac{\lambda}{\lambda + 2}\Sigma_{x|y}.$$

Define $\gamma = \frac{\lambda}{\lambda+2}$, so $\Sigma_{\hat{x}|y} = \gamma\Sigma_{x|y}$. Substituting the latter into the constraint we get

$$\log \left|\left(\gamma\Sigma_{x|y} + \Sigma_{x|y}\right)/2\right| - \frac{1}{2}\log\left|\gamma\Sigma_{x|y}\right| - \frac{1}{2}\log\left|\Sigma_{x|y}\right| = P$$
$$\Rightarrow n \log \frac{1+\gamma}{2} - \frac{n}{2}\log\gamma = P$$
$$\Rightarrow \frac{(1+\gamma)^2}{4\gamma} = e^{\frac{2}{d}P} \tag{28}$$
$$\Rightarrow \gamma^2 + 2\gamma + 1 = 4\gamma e^{\frac{2}{d}P}$$
$$\Rightarrow \gamma(P) = 2e^{\frac{2}{d}P} - 1 - \sqrt{(2e^{\frac{2}{d}P} - 1)^2 - 1}.$$

Thus, we obtain that

$$U_G(P) = \eta(P) \cdot N(X_G|Y) \tag{29}$$

where

$$\eta(P) = \gamma(P) + 1 = 2e^{\frac{2}{d}P} - \sqrt{(2e^{\frac{2}{d}P} - 1)^2 - 1}. \tag{30}$$

Notice that $\eta(0) = 2$, while $\lim_{P\to\infty} \eta(P) = 1$, so $1 \leq \eta(P) \leq 2$. Following similar steps where we replace $\Sigma_{\hat{x}|y}$ and $\Sigma_{x|y}$ with $N(\hat{X}|Y)$ and $N(X|Y)$ respectively, we derive

$$U_g(P) = \eta(P) \cdot N(X|Y). \tag{31}$$

## G  Proof of Theorem 4

Define $\mathcal{E} \triangleq \hat{X} - X$. Then,

$$\frac{1}{d}\mathbb{E}\left[||\hat{X} - X||^2\right] \underset{(a)}{=} \mathbb{E}\left[\frac{1}{d}\mathbb{E}\left[||\hat{X} - X||^2|Y\right]\right] = \mathbb{E}\left[\frac{1}{d}\mathbb{E}\left[||\mathcal{E}||^2|Y\right]\right] = \mathbb{E}\left[\frac{1}{d}\mathbb{E}\left[\mathcal{E}^T\mathcal{E}|Y\right]\right]$$
$$= \mathbb{E}\left[\frac{1}{d}Tr\left(\mathbb{E}\left[\mathcal{E}\mathcal{E}^T|Y\right]\right)\right] = \mathbb{E}\left[\frac{1}{d}Tr\left(\Sigma_{\varepsilon|y}\right)\right]$$
$$\underset{(b)}{\geq} \mathbb{E}\left[\left|\Sigma_{\varepsilon|y}\right|^{1/d}\right] = \mathbb{E}\left[\left|\Sigma_{\hat{x}|y} + \Sigma_{x|y}\right|^{1/d}\right]$$
$$\underset{(c)}{\geq} \mathbb{E}\left[\frac{1}{2\pi e}e^{\frac{2}{d}h(\hat{X} - X|Y=y)}\right]$$
$$\underset{(d)}{\geq} \frac{1}{2\pi e}e^{\frac{2}{d}\mathbb{E}\left[h(\hat{X}-X|Y=y)\right]} = \frac{1}{2\pi e}e^{\frac{2}{d}h(\hat{X}-X|Y)} = N\left(\hat{X} - X|Y\right),$$

where (a) is by the law of total expectation, (b) is due to the inequality of arithmetic and geometric means, (c) follows Lemma 1, and (d) is according to Jensen's inequality.

# H Results via Direct Estimation

Estimating high-dimensional statistics is prone to errors [46]. we used practical measures for perceptual quality and a tractable upper bound for uncertainty. Here, we supplement those results with direct computations of entropy and divergence in a high-dimensional setting. Following prior work [14, 23], we treat images as stationary sources and extract $9 \times 9$ patches. To estimate Rényi divergence for perceptual quality assessment, we first model the probability density functions using kernel density estimation. Subsequently, we compute the divergence through empirical expectations. Uncertainty is estimated using the Kozachenko-Leonenko estimator, which calculates the patch sample differential entropy based on nearest neighbor distances [41, 21, 9, 54]. Results, shown in Figure 8, strongly align with the trends observed in Figure 5.

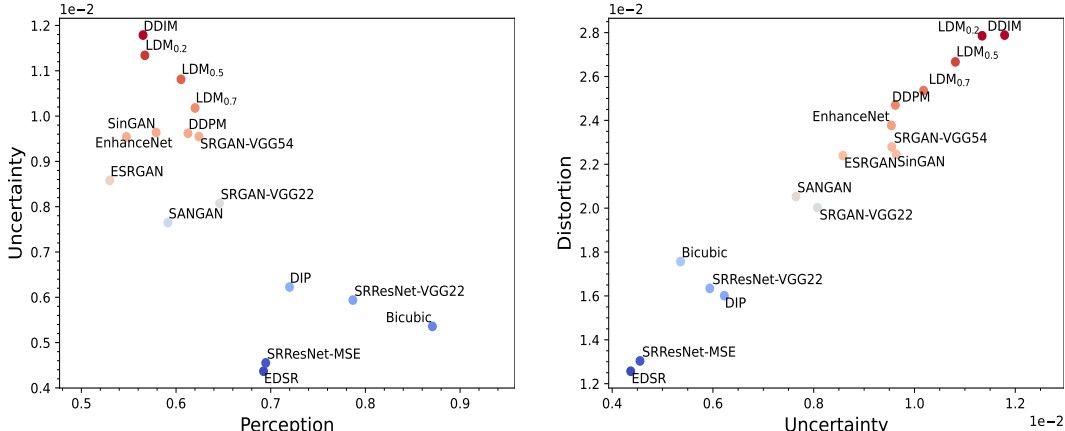

Figure 8: Evaluation of SR algorithms via direct estimation of high-dimensional statistics. Left: Uncertainty-perception plane demonstrating the tradeoff between perceptual quality and uncertainty. Right: Uncertainty-distortion plane illustrating the relation between uncertainty and distortion. Results are consistent with the finding in Figure 5.

