# OpenReview forum: "Looks Too Good To Be True: An Information-Theoretic Analysis of Hallucinations in Generative Restoration Models"
_NeurIPS.cc/2024/Conference — NeurIPS 2024 poster_

### Official Review · Reviewer_DvxH · 2024-06-21

**Soundness:** 3
**Presentation:** 4
**Contribution:** 3
**Rating:** 7
**Confidence:** 3

**Summary:**

This work aims to provide a theoretical analysis about the uncertainty-perception trade-off in generative models, corresponding to the fidelity-naturalness trade-off of the generated images. By defining the inherent uncertainty and formulating a uncertainty-perception （UP) function, the authors proves that the UP function is globally lower-bounded by the inherenet uncertainty. Additionally, they derive that perfect perceptual quality requires at least twice the inherent uncertainty. The proposed theoretical framework establish a relationship between uncertainty and MSE, resembling the well-known perception-distortion trade-off. All theoretical findings are empirically verified with image super-resolution algorithms.

**Strengths:**

+ This is a timely theoretical analysis about the hallucination phenomenon widely occurs in generative models.
+ The proposed theoretical framework helps practitioners better understand the tradeoff between uncertainty and perceptual quality, guiding them to tune the models in real-world safety-sensitive applications.
+ Detailed proofs are provided.

**Weaknesses:**

- In my opinion, LPIPS, MSE, PSNR, SSIM are all full-reference image quality metrics that quantify the fidelity of the restorted images with respect to the ground-truths. Would it be more reasonable to adopt no-reference image quality metrics to quantify the perception ?
- The experiment part is relatively weak, where more quantitative examples are expected.
- The GT image is not presented in Fig. 5, making it difficult to assess the fidelity of the resorted results.

**Questions:**

I think it would be more convincing if the authors can provide more experimental results in real-world SR tasks.

**Limitations:**

Yes.

---

> ### Author Rebuttal · Authors · 2024-08-07
>
> We thank the reviewer for reading our manuscript and providing constructive feedback. Below, we respond to the specific points raised by the reviewer.
>
> **Weaknesses**
>
> 1. While our initial analysis utilized full-reference metrics (LPIPS, MSE, PSNR, SSIM) due to their widespread acceptance, we acknowledge the reviewer's suggestion regarding no-reference metric to quantify perception. We have therefore performed new experiments on image inpainting using state-of-the-art perceptual measures LIQE and Q-ALIGN. The results, available in the 'Author Rebuttal' and accompanying PDF, reinforce our conclusions, demonstrating a clear link between perceptual quality and increased uncertainty. We will revise the manuscript to incorporate these findings.
>
> 2. We acknowledge the reviewer's critique regarding the experimental aspect of our manuscript. In response, we have expanded our evaluation to include latent diffusion models in the context of image inpainting, utilizing the SeeTrue** dataset. This dataset provides a diverse benchmark for image-text alignment, encompassing both real and synthetic text-image pairs. The new results, detailed in the 'Author Rebuttal' and the accompanying PDF, further validate our findings across different restoration tasks and under a wider range of conditions and data distributions.
>
> ** Yarom, M., Bitton, Y., Changpinyo, S., Aharoni, R., Herzig, J., Lang, O., Ofek, E. and Szpektor, I., “What you see is what you read? improving text-image alignment evaluation”. NeurIPS 2023.
>
> 3. We thank the reviewer for highlighting this important point. We will add the ground truth image to the revised manuscript to provide a clearer reference point. Additionally, in response to the reviewer's feedback, we have included new visual results in the 'Author Rebuttal' and the accompanying PDF. These results more effectively illustrate the phenomenon of hallucination in image restoration, clearly demonstrating that the degree of hallucination increases as the perceptual quality of the restored image improves.
>
>
> **Questions**
> 1. We appreciate the reviewer's suggestion to include additional experimental results on real-world super-resolution tasks. While we acknowledge the value of such experiments in demonstrating the real-world applicability of our theoretical findings, time constraints prevented us from conducting them within the scope of this rebuttal. However, we believe that the expanded experiments presented in the 'Author Rebuttal' and accompanying PDF, which include a diverse dataset and the use of no-reference image quality metrics, significantly strengthen the empirical support for our theoretical framework.
> We have included these results to demonstrate the robustness of our findings across different image restoration tasks and under a wider range of conditions and data distributions. We recognize that further validation on real-world SR tasks would be valuable, and we hope to incorporate such experiments in our future work.

---

> > ### Comment · Reviewer_DvxH · 2024-08-08
> > **Post-rebuttal**
> >
> > Thanks for the reponses, I raised my rating to 7.

---

> > > ### Author Response · Authors · 2024-08-11
> > >
> > > We sincerely appreciate the reviewer recognizing our contribution and raising the score accordingly.

---

### Official Review · Reviewer_WuZt · 2024-07-08

**Soundness:** 3
**Presentation:** 3
**Contribution:** 3
**Rating:** 6
**Confidence:** 4

**Summary:**

This paper presents a theoretical perspective towards hallucinations and reveals a tradeoff between uncertainty and perception for image restoration problem. Additionally, the paper points out that uncertainly-perception tradeoff can induce the well-known perception-distortion tradeoff.

**Strengths:**

1. The paper provides a theoretical interpretation about the hallucinations problems of inverse problem, which may offer useful guidance for further practical research.
2. The writing is good and the paper is easy to follow.
3. Rich theoretical results about uncertainly-perception tradeoff and its relationship with perception-distortion tradeoff.

**Weaknesses:**

1. Typically, perception can be measured by criteria like LPIPS.  Why can the conditional convergence (Eq 1) provide measurement for perception? Can the authors provide some intuitive explanation?
2. The theorem 2 is based on a strict assumption that $D_v$ is convex in its second argument. Can you proof this directly?
3. The visual results are not enough. The authors should provide some examples that contain hallucinations. It seems that in Figure 5, the details are realistic-looking details but not hallucinations.

**Questions:**

see weakness

**Limitations:**

The authors discuss the limitations.

---

> ### Author Rebuttal · Authors · 2024-08-07
>
> We appreciate the reviewer's time and effort  in evaluating our manuscript. We have taken the feedback into consideration and responded to the specific points raised below.
>
> **Weaknesses**
>
> 1. In our context, perception is defined as the probability $p_\text{success}$ of a human observer to successfully distinguish between a pair of natural and degraded images (drawn from $p_{X,Y}$) and a pair of restored and degraded images (drawn from $p_{\hat X, Y}$). From a Bayesian perspective, the optimal decision rule maximizing $p_\text{success}$ yields**:
> $$p_\text{success}=\frac{1}{2}+\frac{1}{2}D_\text{TV}(p_{X,Y},p_{\hat X,Y})$$
> where $D_\text{TV}(p_{X,Y},p_{\hat X,Y})$ is the total-variation (TV) distance.
> When $D(p_{X,Y},p_{\hat X,Y})=0$, the two pairs are indistinguishable ($p_\text{success}=0.5$), implying perfect perception quality. We generalize this beyond the total-variation distance to any conditional divergence, recognizing that the divergence that best relates to human perception remains an open question. However, computing divergences in high dimensions is challenging, leading to the use of practical alternatives like LPIPS. Following the reviewer’s comment we will include the above explanation in the revised manuscript.
>
> ** See section 2 in the following paper: ​​Nielsen, F., 2013, August. Hypothesis testing, information divergence and computational geometry. In International Conference on Geometric Science of Information (pp. 241-248). Berlin, Heidelberg: Springer Berlin Heidelberg.
>
> 2. While Theorem 2 assumes convexity of $D_v$ in its second argument, this is not a restrictive condition. In fact, most widely-used divergence functions, notably all $f$-divergences (such as KL divergence, total variation distance, Hellinger distance, and Chi-square divergence), exhibit this property.  This broad applicability is further highlighted by the convexity of the Rényi divergence, used in our work, in its second argument, which we prove next.
> The Renyi divergence with $r\geq 0, r\neq 1$ between two probabilities $p$ and $q$ is given by
> $$D_r(p,q)=\frac{1}{r-1}\int p(x)^rq(x)^{1-r}dx.$$
> Notice that the function $f(q)=\frac{1}{r-1}q^{1-r}$ is convex, therefore, for any $0\leq\lambda\leq 1$ and distributions $q_1$ and $q_2$ we have
> \begin{align*}
>     D_r(p,\lambda q_1+(1-\lambda)q_2)
>     &=\frac{1}{r-1}\int p(x)^r(\lambda q_1(x)+(1-\lambda)q_2(x))^{1-r}dx \\\\
>     &=\int p(x)^r\frac{1}{r-1}(\lambda q_1(x)+(1-\lambda)q_2(x))^{1-r}dx \\\\
>     &\leq \int p(x)^r\frac{1}{r-1}\lambda q_1(x)^{1-r}+ \int p(x)^r\frac{1}{r-1}(1-\lambda) q_2(x)^{1-r} \\\\
>     &=\lambda\frac{1}{r-1} \int p(x)^rq_1(x)^{1-r}+(1-\lambda)\frac{1}{r-1} \int p(x)^rq_2(x)^{1-r} \\\\
>     &= \lambda D_r(p,q_1)+(1-\lambda)D_r(p,q_2),
> \end{align*}
> implying the Renyi divergence is convex.
> Note that the proof above omits the case of $r=1$ for simplicity. For a comprehensive proof and further properties, see Theorem 12 in
> Van Erven, T. and Harremos, P., 2014. Rényi divergence and Kullback-Leibler divergence. IEEE Transactions on Information Theory, 60(7), pp.3797-3820.
>
> 3. We acknowledge the reviewer's concerns regarding the visual results and agree with the suggestion to provide additional examples. To address this, we have conducted further experiments on image inpainting, with new visual results presented in the 'Author Rebuttal' and the accompanying PDF. These results clearly demonstrate the phenomenon of hallucination in image restoration, where the degree of hallucination increases with the improvement in the perceptual quality of the restoration algorithm.

---

> ### Comment · Reviewer_WuZt · 2024-08-09
> **After Rebuttal**
>
> Thanks for the rebuttal. The authors address my questions. I strongly recommend the authors to add more visual results to illustrate the paper clearly. I raise the score to 6.

---

> > ### Author Response · Authors · 2024-08-11
> >
> > We deeply value the reviewer's positive feedback and the subsequent score increase.

---

### Official Review · Reviewer_xyrt · 2024-07-11

**Soundness:** 3
**Presentation:** 3
**Contribution:** 2
**Rating:** 6
**Confidence:** 3

**Summary:**

The paper employs information-theory tools to characterize a tradeoff between uncertainty and perception in image restoration. They prove that high perceptual quality leads to increased uncertainty and the uncertainty-perception trade-off induces the distortion-perception trade-off. The theoretical results are illustrated with experiments in image super-resolution tasks.

**Strengths:**

1. The paper is well-written.
2. The authors provide clear theoretical and quantitative demonstrations.

**Weaknesses:**

1. The trade-off between perception and distortion has been discussed in some previous works. The authors should clarify their contribution especially compare to previous works[1,2].
2. This paper establishes the theoretical relationship between uncertainty and perception. However, the authors do not provide practical applications, e.g. how to use this relationship in restoration task.

Refs:
[1] The Perception-Distortion Tradeoff. CVPR, 2018.
[2] The Perception-Robustness Tradeoff in Deterministic Image Restoration. arXiv:2311.09253.

**Questions:**

1. The uncertainty-perception plane is based on SFID, PDL and LPIPS, the relationship is still exists on stronger vision-language IQA like LIQE[1], Q-ALIGN[2]?
2. Some recent SR methods[3,4] explore a better trade-off between perception and artifacts. How they perform in uncertainty-perception and uncertainty-distortion measurement?

Refs:
[1] Blind image quality assessment via vision-language correspondence: A multitask learning perspective. CVPR, 2023.
[2] Q-ALIGN: Teaching LMMs for Visual Scoring via Discrete Text-Defined Levels. arXiv:2312.17090.
[3] DeSRA: Detect and Delete the Artifacts of GAN-based Real-World Super-Resolution Models. ICML, 2023.
[4] Details or artifacts: A locally discriminative learning approach to realistic image super resolution. CVPR, 2022.

**Limitations:**

See weakness

---

> ### Author Rebuttal · Authors · 2024-08-06
>
> We thank the reviewer for their insightful comments and constructive feedback. Below we address in detail the major concerns raised by the reviewer.
>
> **Weaknesses**
>
> 1. We acknowledge the extensive literature on the perception-distortion tradeoff, particularly the seminal work demonstrating the inherent conflict between these two factors [1]. This work quantifies perception using a theoretical divergence between true and estimated distributions, establishing a tradeoff valid for any distortion measure. More recent research [2] has introduced a tradeoff between perception and robustness in restoration algorithms, utilizing the Lipschitz constant as a measure of robustness and defining "joint perceptual quality" based on the Wasserstein distance between joint distributions. This study reveals that improving joint perception leads to increased sensitivity in algorithms.
>
> While these works provide valuable insights into the relationship between perceptual quality, distortion, and robustness, crucial aspects for understanding generative models, our work offers a distinct contribution by focusing on the concept of uncertainty. Although related, uncertainty plays a different role than distortion in image restoration. Distortion metrics quantify the fidelity of the restored image to the ground truth, typically through error size. Uncertainty metrics, on the other hand, quantify the range of possible solutions, characterizing confidence in the restoration itself. This distinction is vital for decision-making, as high distortion may lead to incorrect choices, while high uncertainty implies low confidence, making it difficult to make any informed decision. Our work thus complements existing research by incorporating uncertainty into the broader discussion of image restoration quality.
>
> In the revised manuscript, we will expand the discussion on these references [1, 2] to further emphasize our unique contribution and the points mentioned above.
>
> 2. We understand the reviewer's concerns regarding the practical implications of our work. Our primary goal is to raise awareness among developers in various fields about the inherent tradeoff between perception and uncertainty. This understanding allows them to prioritize safety and reliability alongside perceptual enhancements when integrating cutting-edge models into desired applications.
> In practice, our proposed uncertainty-perception plane can serve as a valuable tool for evaluating potential restoration algorithms, facilitating the identification of methods that achieve the desired balance for specific applications. Moreover, the tractable uncertainty measure used in our experiments, or any differentiable alternative, can be incorporated into a loss function during the training of generative models like GANs or as an optimization objective to guide the reverse process in diffusion models. This approach enables the development of algorithms that explicitly optimize for the tradeoff between uncertainty and perception.
>
> In the revised manuscript, we will expand the discussion on the practical implications of our work on the development and evaluation of image restoration models.
>
> **Questions**
>
> 1. We thank the reviewer for raising this important point. While we initially used SFID, PDL, and LPIPS as common and widely accepted measures of perception, we acknowledge the significant progress in developing new metrics like LIQE and Q-ALIGN that better align with human perception. In response to the reviewer’s feedback, we have conducted additional experiments on image inpainting, measuring perception with LIQE and Q-ALIGN. The results, provided in the 'Author Rebuttal' and accompanying PDF, support our existing findings, showing that an increase in perceptual quality generally correlates with increased uncertainty. We will include these additional experiments in the revised manuscript.
>
> 2. We appreciate the reviewer highlighting the recent advancements in super-resolution (SR) methods that explore the trade-off between perception and artifacts [3, 4]. While time constraints prevented us from evaluating these specific methods within the rebuttal period, we are confident that our theoretical analysis remains valid regardless of the specific recovery algorithm. Our framework establishes a fundamental relationship between uncertainty and perception, independent of the algorithm's implementation details. Therefore, we anticipate that the mentioned SR methods would exhibit behavior consistent with our findings, demonstrating a similar relationship between uncertainty and perception. In the revised manuscript, we plan to include an evaluation of these recent SR methods to further validate and expand upon our theoretical framework.

---

> > ### Comment · Reviewer_xyrt · 2024-08-12
> > **After Rebuttal**
> >
> > Thank you for the detailed response. I'll keep my score.

---

> > > ### Author Response · Authors · 2024-08-12
> > >
> > > We sincerely thank the reviewer for their valuable feedback, particularly the request to incorporate non-reference perceptual measures, which has significantly strengthened our contribution.

---

### Official Review · Reviewer_W9yk · 2024-07-12

**Soundness:** 3
**Presentation:** 3
**Contribution:** 3
**Rating:** 6
**Confidence:** 4

**Summary:**

Deep generative models have achieved remarkable performance in image restoration, resulting in generated images of high visual quality. However, these models often produce high-frequency details that are not consistent with the ground-truth images. Such hallucinations introduce uncertainty in the generated content and affect the reliability of model predictions. This paper defines the uncertainty of image restoration and the uncertainty-perception (UP) function, and reveals an uncertainly-perceptual trade-off. The paper theoretically analysis the relationship between the uncertainly-perceptual trade-off and the perceptual-distortion trade-off. The theoretical findings are validated through experiments with image super-resolution algorithms. Results show that no model can achieve both low uncertainty and high perceptual quality simultaneously.

**Strengths:**

1.  This is a timely paper that analyses the hallucination of deep generative models. It proposes a novel insight of the phenomenon of hallucinations in generative models, a critical issue that affects the reliability of image restoration tasks.

2. The paper adopts the Bayesian framework to analyze the tradeoff between uncertainty and perception. This framework helps in quantifying the inherent uncertainty in generative models and establishes a theoretical foundation for understanding the limitations of these models.

3. The theoretical findings are empirically validated using single-image super-resolution algorithms. This strengthens the credibility of the theoretical analysis of the study.

4. Following the definition, a concrete example (e.g., example 1) is illustrated which can help the readers better understand the concept.

**Weaknesses:**

1. Experiments only provide results in image super-resolution, how about applying the proposed method on other restoration tasks?
2. The experiments primarily use synthesized datasets, such as BSD100. The paper would benefit from including experiments on more diverse and real-world datasets to validate the findings under different conditions and data distributions.
3. The paper relies on entropy and Rényi divergence for theoretical analysis. However, the practical estimation of high-dimensional entropy is challenging. Although the authors use a tractable upper bound for uncertainty, the practical estimation methods and their limitations are not thoroughly discussed

**Questions:**

1. The paper [1] quantifies the structural uncertainty for image restoration. The intuitive explanation is better for readers to understand the inherent uncertainty mentioned in the paper.

2. The practical estimation for computing divergence and uncertainty should be elaborated, which is challenging for real-world images that are typically high-dimensional.

3. While the theoretical analysis is robust, the paper could benefit from more concrete examples of how the findings apply to real-world scenarios, like healthcare or autonomous systems.

[1] Belhasin O, Romano Y, Freedman D, Rivlin E, Elad M. Principal uncertainty quantification with spatial correlation for image restoration problems. IEEE Transactions on Pattern Analysis and Machine Intelligence. 2023 Dec 14.

**Limitations:**

The paper aims to quantify the potential limitation (e.g., hallucination) of generative models. At the end of the paper, it discusses the limitations of the proposed method.

---

> ### Author Rebuttal · Authors · 2024-08-06
>
> We appreciate the time and effort the reviewer has dedicated to reviewing our manuscript. Below we address in detail the major concerns raised by the reviewer.
>
> **Weaknesses**
>
> 1-2. *Experiments* - We acknowledge the reviewer's concerns regarding the experiments on image super-resolution using the BSD100 dataset. While our primary focus is theoretical, we sought to validate our findings via numerical experiments. We initially chose super-resolution due to its status as a benchmark in image restoration, the wide availability of datasets and models, and its extensive study in related literature on the perception-distortion tradeoff. However, in response to the reviewer's criticism, we have expanded our experiments to include an evaluation of latent diffusion models in image inpainting using the SeeTrue* dataset, a diverse benchmark for image-text alignment containing both real and synthetic text-image pairs. These new results, detailed in the 'Author Rebuttal' and the accompanying PDF, align with our existing findings, showing their validity across different restoration tasks and under diverse conditions and data distributions.
>
> *Yarom, M., Bitton, Y., Changpinyo, S., Aharoni, R., Herzig, J., Lang, O., Ofek, E. and Szpektor, I., “*What you see is what you read? improving text-image alignment evaluation*”. NeurIPS 2023.
>
> 3. Our work primarily focuses on the theoretical analysis of the relationship between uncertainty and perception, leveraging the properties of information-theoretic measures like entropy and divergence. However, we acknowledge the reviewer's valid concern regarding the challenges of estimating these statistics in high dimensions. In response, we have expanded our discussion on this topic in the revised manuscript to include a comprehensive review of practical estimation methods and their limitations. The following discussion, along with the accompanying full references, will be included in the revised manuscript:
>
> Information theory offers a powerful framework for quantifying uncertainty and dependencies in data, handling multivariate and heterogeneous data types, and capturing complex patterns. However, its wider adoption has been limited by the challenge of estimating information-theoretic measures in high dimensions. The curse of dimensionality makes accurate density estimation infeasible [1, 2], leading many to rely on simpler second-order statistics.
>
> The development of practical tools for estimating statistics in high-dimensional data remains an active area of research [3]. While initial approaches assumed exponential family distributions (e.g., Gaussian) for tractable calculations [4], their performance degrades for long-tailed distributions. Non-parametric methods like binning strategies, including KDE and kNN estimators [5, 6, 7], offer more flexibility but are data-dependent and sensitive to parameter choices. Alternative approaches involve ensemble estimation [8] or von Mises Expansions [9], the distributional analog of the Taylor expansion. Rotation-Based Iterative Gaussianization (RBIG) [10] presents a promising direction by transforming data into a multivariate Gaussian domain, simplifying density estimation. However, its application to images has been limited to small patches due to the computational challenges of learning rotations based on principal or independent component analysis. A recent extension addresses this by utilizing convolutional rotations, enabling efficient processing of entire images [11].
>
> Our theoretical study investigates the relationship between uncertainty, perception, and distortion in image restoration through the lens of information theory. We uncover a novel uncertainty-perception tradeoff and its connection to the well-known distortion-perception tradeoff. While primarily theoretical, our analysis yields a practical measure of uncertainty (or entropy), used to visually and quantitatively illustrate our findings. This measure potentially can be replaced by the aforementioned estimators for broader applications.
>
>
> **Questions**
>
> 1. Our work utilizes the Bayesian framework, treating recovery error as a random variable and employing entropy as a natural measure of uncertainty**. However, estimating entropy in high dimensions presents challenges. Over time, practical methods have emerged that rely on alternative definitions of uncertainty. One such method, proposed by Belhasin et al., quantifies uncertainty volume using principal components of the empirical posterior probability density function. We agree with the reviewer that this novel definition is intuitive and leads to a practical method for uncertainty quantification, which we briefly discuss in the current manuscript. In the revised version, we will expand our discussion and description of this approach, acknowledging its merits and relevance to our work.
>
> ** Cover, Thomas M.; Thomas, Joy A. (1991). *Elements of Information Theory*.
>
> 2. Please see the responses to weakness above. We have carefully considered your feedback and, in response, have expanded our discussion on advancements in the practical estimation of statistics in high dimensions.
>
> 3. Developers across various fields, including healthcare and autonomous systems, often integrate cutting-edge models into their applications, prioritizing state-of-the-art performance and perceptual quality. However, our work aims to highlight a crucial factor often overlooked: the inherent tradeoff between uncertainty and perception. By raising awareness of this tradeoff, we empower developers to make informed decisions that prioritize safety and reliability over purely perceptual enhancements. For instance, in healthcare, potential restoration algorithms can be evaluated by plotting them on the uncertainty-perception plane, facilitating the identification of methods that strike the optimal balance for specific clinical needs. In light of your feedback, we will address the points mentioned above in the revised manuscript.

---

> > ### Comment · Reviewer_W9yk · 2024-08-11
> >
> > Thank you for your responses and comprehensive explanation. Your responses have addressed mu concerns, and I will decide to raise the score to WA.

---

> > > ### Author Response · Authors · 2024-08-11
> > >
> > > We are thankful for the reviewer's insightful comments and the resulting score change.

---

### Author Rebuttal · Authors · 2024-08-07

We sincerely thank the reviewers for their efforts in evaluating our manuscript, their overall positive feedback, and their constructive criticisms, which have significantly strengthened our contribution. We have provided detailed responses to each reviewer individually. In the following, we summarize the major revisions made in response to the collective feedback:

**Extended Experiments**

* Image Inpainting: In addition to our initial super-resolution experiments, we have conducted extensive new experiments on image inpainting. We specifically chose latent diffusion models for this task due to their state-of-the-art performance and growing popularity.
* Diverse Dataset: We utilized the SeeTrue* dataset, a benchmark known for its diverse range of real and synthetic text-image pairs. This allows us to assess the validity of our findings across different restoration tasks and under a broader spectrum of conditions and data distributions.

*Yarom, M., Bitton, Y., Changpinyo, S., Aharoni, R., Herzig, J., Lang, O., Ofek, E. and Szpektor, I., “What you see is what you read? improving text-image alignment evaluation”. NeurIPS 2023.

* No-Reference Metrics: We incorporated the state-of-the-art no-reference perceptual quality metrics LIQE and Q-ALIGN to quantify perception in these new experiments.

* Visual and Quantitative Results: We have included new visual and quantitative results in the attached PDF. These results support our existing findings, showing that an increase in perceptual quality generally correlates with increased uncertainty.  Furthermore, the visual results more effectively illustrate the phenomenon of hallucination in image restoration, clearly demonstrating the increase in hallucination with improved perceptual quality.

**Extended Discussion and Theoretical Context**

* Practical Implications: We have expanded the discussion on the practical implications of our theoretical analysis, emphasizing how our findings can guide the development and evaluation of image restoration models, particularly in safety-critical applications.
* Estimation of High-Dimensional Statistics: We have provided a more comprehensive discussion on the challenges and advancements in estimating statistics like entropy and divergence in high dimensions. This includes an overview of practical estimators and their limitations, offering valuable insights for researchers and practitioners.
* Related Work: We have further clarified our unique contribution by expanding the discussion on related previous works and highlighting the distinct role that uncertainty plays in image restoration compared to distortion or robustness.

We are confident that these revisions address the reviewers' concerns and enhance the overall quality and impact of our work. We once again thank the reviewers for their valuable feedback and hope that our responses adequately address their questions and suggestions.

---

### Decision · Program_Chairs · 2024-09-25

**Decision:**

Accept (poster)

**Comment:**

This paper provides a theoretical analysis of the tradeoff between perceptual quality and uncertainty in generative image restoration models, revealing that higher perceptual quality leads to increased uncertainty, often manifesting as hallucinations. The work is grounded in information theory, with the authors establishing a connection between the uncertainty-perception tradeoff and the perception-distortion tradeoff. The theoretical findings are empirically validated through experiments, initially focused on single-image super-resolution.

Reviewers appreciated the paper's theoretical rigor and its timely contribution to understanding the limitations of generative models. However, concerns were raised about the limited scope of the experiments and the practical applicability of the findings. In response, the authors extended their experiments to include image inpainting and employed no-reference perceptual quality metrics, which addressed these concerns effectively. They also provided a more comprehensive discussion on the challenges of estimating high-dimensional statistics and added additional visual results to better illustrate the phenomenon of hallucinations.

Overall, the paper was well-received after the discussions, with reviewers acknowledging the strengthened empirical support and the practical relevance of the work. The recommendation is to accept the paper as a poster presentation, given its solid theoretical contribution and the effective responses to reviewer concerns.